# Taxonomy and Phylogeny of Cystostereaceae (Agaricales, Basidiomycota): A New Genus, Five New Species, and Three New Combinations

**DOI:** 10.3390/jof8111229

**Published:** 2022-11-21

**Authors:** Yue Li, Karen K. Nakasone, Che-Chih Chen, Chang-Lin Zhao, Ting Cao, Hai-Sheng Yuan, Shuang-Hui He

**Affiliations:** 1School of Ecology and Nature Conservation, Beijing Forestry University, Beijing 100083, China; 2Center for Forest Mycology Research, Northern Research Station, U.S. Forest Service, Madison, WI 53726, USA; 3Department of Biology, National Museum of Natural Science, Taichung 404023, Taiwan; 4College of Biodiversity Conservation, Southwest Forestry University, Kunming 650224, China; 5School of Public Health, Shengyang Medical College, Shenyang 110034, China; 6CAS Key Laboratory of Forest Ecology and Management, Institute of Applied Ecology, Chinese Academy of Sciences, Shenyang 110164, China

**Keywords:** corticioid fungi, *Crustomyces*, *Cystostereum*, *Parvodontia*, white rot, wood-decaying fungi

## Abstract

This paper aims to understand the species diversity, taxonomy, and phylogeny of Cystostereaceae (Agaricales), which is based primarily on material from East and Southeast Asia. Cystostereaceae is a small, understudied family of saprobes of woody plants with a worldwide distribution. Phylogenetic analyses of the LSU and ITS sequences revealed four distinct clades in the Cystostereaceae, representing the genera *Crustomyces*, *Cystostereum*, *Effusomyces* gen. nov., and *Parvodontia*. In addition, phylogenetic analyses showed that *Cystidiodontia* and *Rigidotubus* are synonyms of *Crustomyces* for their type of species nested within the *Crustomyces* clade. The new monotypic genus *Effusomyces*, based on specimens from Thailand, lacks any distinctive morphological features. *Parvodontia*, originally erected for a species from South America, is reported for the first time from Asia. The widely distributed and morphologically well-characterized *Cystostereum* is represented in East Asia by two new species: *Cystostereum crassisporum* and *C. submurrayi*. In addition, three new species, viz., *Crustomyces albidus*, *Effusomyces thailandicus*, and *Parvodontia austrosinensis*, are described and illustrated. Finally, three new combinations are proposed: *Crustomyces isabellinus*, *C. laminiferus*, and *C. tephroleucus*. A key to the genera and species of Cystostereaceae is provided. Our results proved that the species diversity of wood-decaying fungi in East and Southeast Asia is rich and suggested that more investigations and studies should be carried out in the future.

## 1. Introduction

Cystostereaceae Jülich is a small family in the Agaricales of wood-decaying fungi composed of seven genera [1,2]. Established in 1982, it included two genera —*Cystostereum* Pouzar and *Crustomyces* Jülich [1,3,4]. Later, *Cericium* Hjortstam, *Cystidiodontia* Hjortstam, *Parvobasidium* Jülich, *Parvodontia* Hjortstam & Ryvarden, and *Rigidotubus* J. Song, Y.C. Dai & B.K. Cui was included in the family [2,5,6]. Except for *Rigidotubus*, the genera were created based on morphological characters alone. There are only 20 species in the family listed by Index Fungorum [https://www.indexfungorum.org (accessed on 1 June 2022)] and MycoBank [https://www.mycobank.org (accessed on 1 June 2022)], and *Cystostereum* is the largest genus with eight species. *Cericium* and *Rigidotubus* are monotypic, *Parvobasidium* and *Parvodontia* have two species each, and *Crustomyces* and *Cystidiodontia* have three.

Basidiomes in the Cystostereaceae are effused to effused-reflexed with smooth, odontoid, hydnoid, or poroid hymenophores with a monomitic or dimitic hyphal system, clamped generative hyphae, gloeocystidia or hyphidia, and smooth basidiospores that do not react in Melzer’s reagent. Microbinding and skeletal hyphae are developed in a few species. In species with gloeocystidia, the contents are typically dark yellow, resinous-like, and do not react with sulfovanillin. Ecologically, the species are saprobes that inhabit logs and branches of conifers or hardwoods, causing a white rot [1,2]. Unfortunately, only a few species in the Cystostereaceae have sequence data available, so phylogenetic relationships among the genera and species are preliminary and incomplete [2,5].

In this study, we examine the taxonomy and phylogeny of taxa in the Cystostereaceae, emphasizing East and Southeast Asian species. We focus on species in *Crustomyces*, *Cystidiodontia*, *Cystostereum*, *Parvodontia*, and *Rigidotubus*, because sequence data for the other genera are unavailable. The first sequence data for *Parvodontia*, based on a newly described taxon from southern China, is presented. From phylogenetic analyses, we determine that *Rigidotubus* and *Cystidiodontia* are synonyms of *Crustomyces*. Molecular and morphological data were used to establish the new genus *Effusomyces* as well as five new species. All new taxa are described and illustrated, and a key to the genera and species of the Cystostereaceae is provided.

## 2. Materials and Methods

### 2.1. Specimen Collection

In situ photos of specimens were taken with a Canon camera EOS 70D (Canon Corporation, Tokyo, Japan). Specimens were dried with a portable dryer, labeled, then stored in a freezer at minus 40 °C for two weeks to kill the insects and their eggs before proceeding with morphological and molecular studies. Voucher specimens are deposited at the herbaria of Beijing Forestry University, Beijing, China (BJFC), Institute of Applied Ecology, Academia Sinica, Shenyang, China (IFP), Southwest Forestry University, Kunming, China (SWFC), National Museum of Natural Science, Taichung, Taiwan, China (TNM), and Center for Forest Mycology Research, Madison, WI, USA (CFMR). Herbarium code designations follow Index Herbarium [http://sweetgum.nybg.org/science/ih/(accessed on 1 August 2022)].

### 2.2. Morphological Studies

Thin, freehand sections were made from dried basidiomes and mounted in 2% (weight/volume) aqueous potassium hydroxide (KOH) and 1% (*w*/*v*) aqueous phloxine. Amyloidity and dextrinoidity of hyphae and basidiospores were checked in Melzer’s reagent (IKI). Cyanophily of hyphal and basidiospore walls were observed in 1% (*w*/*v*) cotton blue in 60% (*w*/*v*) lactic acid (CB). Microscopic examinations were carried out with a Nikon Eclipse 80i microscope (Nikon Corporation, Tokyo, Japan) at magnifications up to 1000×. Drawings were made with the aid of a drawing tube. The following abbreviations are used: IKI– = neither amyloid nor dextrinoid, CB+ = cyanophilous, CB– = acyanophilous, L = mean spore length, W = mean spore width, Q = L/W ratio, and n (a/b) = number of spores (a) measured from the number of specimens (b). Color codes and names follow Kornerup and Wanscher [7].

### 2.3. DNA Extraction and Sequencing

A CTAB plant genomic DNA extraction kit, DN14 (Aidlab Biotechnologies Co., Ltd., Beijing, China), was used to extract total genomic DNA from dried specimens, amplified by the polymerase chain reaction (PCR), according to the manufacturer’s instructions. The ITS1-5.8S-ITS2 region was amplified with the primer pair ITS5/ITS4 [8] using the following protocol: initial denaturation at 95 °C for 4 min, followed by 34 cycles at 94 °C for 40 s, 58 °C for 45 s and 72 °C for 1 min, and final extension at 72 °C for 10 min. The D1-D2 region of the nucleic ribosomal LSU was amplified with the primer pair LR0R/LR7 [http://www.biology.duke.edu/fungi/mycolab/primers.htm (accessed on 1 August 2022)] employing the following procedure: initial denaturation at 94 °C for 1 min, followed by 34 cycles at 94 °C for 30 s, 50 °C for 1 min and 72 °C for 1.5 min, and final extension at 72 °C for 10 min. DNA sequencing was performed at the Beijing Genomics Institute, and newly generated sequences were deposited in GenBank [https://www.ncbi.nlm.nih.gov/(accessed on 1 August 2022)]. BioEdit v.7.0.5.3 [9] and Geneious Basic v.11.1.15 [10] were used to review the chromatograms and for contig assembly.

### 2.4. Phylogenetic Analyses

Table 1 lists the taxa and sequences used in the phylogenetic analyses. Two datasets, the LSU sequences of representative taxa of Agaricales and the concatenated ITS-LSU sequences of species of the Cystostereaceae, were constructed. *Coniophora olivacea* (Fr.) P. Karst. and *C. puteana* (Schumach.) P. Karst. were selected as the outgroup for the LSU dataset, whereas *Chondrostereum purpureum* (Pers.) Pouzar was used in the ITS-LSU dataset [2,5]. For the latter dataset, sequences of ITS and LSU were aligned separately using MAFFT v.7 [http://mafft.cbrc.jp/alignment/server/(accessed on 1 August 2022)] [11] with the G-INS-I iterative refinement algorithm and optimized manually in BioEdit v.7.0.5.3. The separate alignments were then concatenated using Mesquite v.3.5.1 [12].

Maximum parsimony (MP), maximum likelihood (ML) analyses, and Bayesian inference (BI) were carried out by using PAUP* v.4.0b10 [13], RAxML v.8.2.10 [14] and MrBayes 3.2.6 [15], respectively. In MP analysis, trees were generated using 100 replicates of random stepwise addition of sequence and the tree-bisection reconnection (TBR) branch-swapping algorithm with all characters given equal weight. Branch supports for all parsimony analyses were estimated by performing 1000 bootstrap replicates with a heuristic search of 10 random-addition replicates for each bootstrap replicate. In ML analysis, statistical support values were obtained using rapid bootstrapping with 1000 replicates, with default settings for other parameters. For BI, the best-fit substitution model was estimated with jModeltest v.2.17 [16]. Four Markov chains were run for two million for the Agaricales LSU and Cystostereaceae ITS-LSU datasets; until the split deviation frequency value was lower than 0.01. Trees were sampled every 100th generation. The first quarter of the trees, which represented the burn-in phase of the analyses, were discarded. The remaining trees were used to calculate posterior probabilities (BPP) in the majority rule consensus tree.

## 3. Results

### 3.1. Phylogenetic Analyses

The LSU dataset consisted of 45 samples representing 31 taxa in the Agaricales, representing seven families with crust or effused basidiomes and the outgroup (Table 1). In this dataset, the aligned length was 1374 characters, of which 276 were parsimony informative. MP analysis yielded eight equally parsimonious trees (TL = 1119, CI = 0.512, RI = 0.757, RC = 0.388, HI = 0.488). The concatenated ITS-LSU dataset comprised 28 ITS and 26 LSU sequences from 35 samples representing 13 taxa in the Cystostereaceae and the outgroup (Table 1). This dataset had an aligned length of 1989 characters, and 277 characters were parsimony informative. MP analysis yielded two equally parsimonious tree (TL = 790, CI = 0.730, RI = 0.839, RC = 0.613, HI = 0.270). jModelTest suggested GTR + I+G was the best-fit model of nucleotide evolution for both datasets. The average standard deviation of the split frequencies of BI was 0.003928 (for the LSU dataset) and 0.003985 (for the ITS-LSU) at the end of the run. ML and BI analyses resulted in almost identical tree topologies compared to the MP analysis for both datasets. The ML trees of Agaricales and Cystostereaceae are shown in Figure 1 and Figure 2, respectively, with the parsimony bootstrap values (≥50%, first value), likelihood bootstrap values (≥50%, second value) and Bayesian posterior probabilities (≥0.95, third value) labeled along the branches.

In Figure 1, seven families were recognized in the Agaricales, confirming the results presented by Song et al. [5], with the Cystostereaceae clade receiving strong support (97/94/1). Four subclades (lineages) corresponding to the genera *Crustomyces*, *Cystostereum*, *Effusomyces* gen. nov., and *Parvodontia* were strongly supported within the Cystostereaceae in both phylogenetic trees (Figure 1 and Figure 2). Significantly, the type species of *Cystidiodontia* (represented by *Crustomyces laminifera*) and *Rigidotubus*, *R. tephroleucus*, nested within the *Crustomyces* clade. In addition, five strongly supported lineages were recovered, representing the new species *Crustomyces albidus*, *Cystostereum submurrayi*, *C. crassisporum*, *Effusomyces thailandicus*, and *Parvodontia austrosinensis* (Figure 2).

### 3.2. Taxonomy

***Crustomyces*** Jülich Persoonia 10:140. 1978, *emended*

Type species: *Crustomyces subabruptus* (Bourdot & Galzin) Jülich

=*Cystidiodontia* Hjortstam, Mycotaxon 17: 571, 1983. [MB#25796]

=*Rigidotubus* J. Song, Y.C. Dai & B.K. Cui, Phytotaxa 333(2): 263, 2018. [MB#823702]

Basidiomes annual or perennial, resupinate, widely effused, adnate, ceraceous to crustaceous, rarely membranous, lacking hyphal cords or rhizomorphs. Hymenophore; smooth, papillate to spinose or poroid, white, cream, gray to grayish brown. Hyphal system; mono- or dimitic; generative hyphae clamped and skeletal hyphae, branched or not, sometimes dextrinoid. Gloeocystidia; usually present, empty, or containing dark yellow material. Dendrohyphidia; present or absent. Basidia clavate; up to 25 μm long, thin-walled, colorless, with four sterigmata and a basal clamp connection. Basidiospores; subglobose, ellipsoid, short-cylindrical, thin- or thick-walled, colorless, smooth, IKI–, CB–, or CB+ if thick-walled.

Notes—The generic concept of *Crustomyces* is revised to include species with monomitic hyphal systems, poroid hymenophores, and thick-walled basidiospores. Originally, two species were included in the genus, *C. subabruptus* (Bourd. & Galzin) Jülich, the generic type, and *C. pini-candensis* (Schwein.) Jülich [4]. Phylogenetic analyses show that the generic type of *Cystidiodontia*, *Hydnum artocreas* Berk. & M.A. Curtis ex Cooke which is a later synonym of *H. laminiferum* Berk. & M.A. Curtis, and *Rigidotubus*, *R. tephroleucus*, were included with *C. subabruptus* in a strongly supported clade, demonstrating that they are congeneric (Figure 1 and Figure 2).

As revised, we accept seven species in *Crustomyces*, the two original species as well as *C. indecorus* Hjortstam, the new taxon *C. albidus* described below, and three new combinations proposed below—*C. isabellinus*, *C. laminiferum*, and *C. tephroleucus*. *Crustomyces expallens* (Bres.) Hjortstam is excluded from *Crustomyces* because its narrowly clavate basidia, 30–40 × 5–7 μm, are decidedly different from the typical, shorter clavate basidia found in other species in the genus.

***Crustomyces albidus*** Yue Li, Nakasone, C.L. Zhao & S.H. He, **sp. nov.** Figure 3

MycoBank: MB844198

Type—China, Yunnan Province, Xinping County, Mopanshan Forest Park, on dead angiosperm branches, 10 November 2019, He 6164 (BJFC 033109, holotype).

Etymology refers to the color of the basidiomes.

Fruiting body—Basidiomes annual, resupinate, widely effused, adnate, inseparable from substrate, coriaceous, first as small patches, later confluent up to 10 cm long, 3.5 cm wide, up to 80 µm thick in section. Hymenophore; smooth, white (3A1) to yellowish white (3A2), turning black in KOH, not cracked; margin thinning out, adnate, indistinct, fimbriate, concolorous with hymenophore surface. Context white.

Microscopic structures—Hyphal system dimitic; generative hyphae bearing clamp connections, microbinding hyphae frequently present near or in the substrate. Subiculum distinct; up to 120 µm thick, an interwoven, slightly compact tissue. Generative hyphae; thin- to slightly thick-walled, colorless, smooth, frequently branched and septate, 2.5–5 µm in diameter; microbinding hyphae slightly thick-walled, colorless, smooth, flexuous, moderately branched, 1–1.5 µm in diameter Subhymenium thin; hyphae thin-walled, colorless, smooth, moderately branched and septate, slightly interwoven, 1.5–3 µm in diameter Gloeocystidia originated from subhymenium, moniliform, sometimes with lateral appendages, empty or with hyaline contents, thin-walled, colorless, smooth, with a basal clamp, not projecting, 30–55 × 6–10 µm. Basidia clavate; thin-walled, colorless, smooth, with four sterigmata and a basal clamp connection, 14–22 × 4–5.5 µm. Basidiospores are short-cylindrical, with an apiculus, thin-walled, colorless, smooth, IKI–, CB–, 4.2–6 × 2–3 µm, L = 5 µm, W = 2.3 µm, Q = 2.1–2.3 (n = 90/3).

Additional specimens examined—China, Yunnan Province, Chuxiong, Zixishan Forest Park, on a fallen angiosperm branch, 1 July 2021, CLZhao 21086 (BJFC, SWFC); 2 July 2021, CLZhao 21168 (BJFC, SWFC); Pu’er, Jingdong County, Wuliangshan Nature Reserve, on a fallen angiosperm branch, 5 October 2017, CLZhao 4176 (BJFC, SWFC); Yuxi, Xinping County, Mopanshan Forest Park, on a fallen angiosperm branch, 18 January 2018, CLZhao 6243 (BJFC, SWFC) & CLZhao 6275 (BJFC, SWFC); on an angiosperm stump, 18 January 2018, CLZhao 6269 (BJFC, SWFC).

Habitat and distribution—on an angiosperm branch or stump causing white rot; known from the Yunnan Province, southwestern China.

Notes—*Crustomyces albidus* is easily identified because of the moniliform gloeocystidia. It is found in the Yunnan Province, southwestern China, where it is locally common. In Figure 1 and Figure 2, *C. albidus* is sister to *C. subabruptus*, a north temperate species from North America and Europe that is characterized by a gray to pale ochraceous basidiome with densely crowded aculei, a dimitic hyphal system with skeletal hyphae, often with dendrohyphidia, and shorter basidiospores, 3.5–4.5 µm long [24].

***Crustomyces isabellinus*** (Berk. & Broome) Yue Li, Nakasone & S.H. He, **combinatio nova** Figure 4MycoBank: MB844199

≡*Kneiffia isabellina* Berk. & Broome, Journal of the Linnean Society. Botany 14: 62, 1875. [MB 230180]

≡*Cystidiodontia isabellina* (Berk. & Broome) Hjortstam & Ryvarden, Mycotaxon 25 (2): 549, 1986. [MB103254]

=*Hypochnicium grandinioides* Ryvarden, Bulletin du Jardin Botanique National de Belgique 48 (1–2): 91, 1978. [MB315573]

Descriptions and illustrations—Hallenberg and Ryvarden [25], as *Cystostereum artocreas*; Ryvarden [26], as *H. grandinioides*; Niemelä et al. [27].

Specimens examined—China, Guangxi Autonomous region, Huanjiang County, Mulun Nature Reserve, on fallen angiosperm trunk, 10 July 2017, He 4755 (BJFC 024274, CFMR); Jinxiu County, Dayaoshan Nature Reserve, Shengtang Mountains, on fallen angiosperm trunk, 15 July 2017, He 4852 (BJFC 024371) & He 4872 (BJFC 024391); Yinshan Park, on dead angiosperm trunk, 16 July 2017, He 4884 (BJFC 024403); Longzhou County, Nonggang Nature Reserve, on fallen angiosperm twig, 20 July 2012, He 20120720-6 (BJFC 014478) & He 20120720-7 (BJFC 014479); Tianlin County, Cenwanglaoshan Nature Reserve, on dead angiosperm branch, 18 July 2012, He 20120718-6 (BJFC 014471); Xing’an County, Mao’ershan Nature Reserve, on fallen angiosperm trunk, 21 August 2011, Yuan 5767 (IFP 019380); Guizhou Province, Libo County, Maolan Nature Reserve, on fallen angiosperm trunk, 11 July 2017, He 4766 (BJFC 024284); Hainan Province, Baoting County, Qixianling Forest Park, on dead angiosperm branch, 18 March 2016, He 3582 (BJFC 022082, CFMR); on fallen angiosperm trunk, 1 July 2019, He 5955 (BJFC 030831). Vietnam, Lam Dong Province, Lac Duong District, Thac Mai Preservation Park, on dead angiosperm tree, 14 October 2017, Yuan 12642 (IFP 019381) & Yuan 12644 (IFP 019382).

Habitat and distribution—on woody angiosperms from Asia and Africa.

Notes—*Kneiffia isabellina* was described from Sri Lanka and later reported from Taiwan and Africa [27,28]. Our specimens from mainland China and Vietnam have grandinioid to odontoid hymenophores and strongly dextrinoid tissues. The closely related *Crustomyces laminiferus*, discussed below, has a South American and Caribbean distribution [27,29]. The phylogenetic trees in Figure 1 and Figure 2 confirm that *C. isabellinus* samples from Asia formed a distinct lineage sister to *C. laminiferus* samples from Costa Rica and Nigeria with strong support values.

***Crustomyces laminiferus*** (Berk. & M.A. Curtis) Yue Li, Nakasone & S.H. He,
**combinatio nova**


MycoBank: MB844200

≡*Hydnum laminiferum* Berk. & M.A. Curtis, Journal of the Linnean Society. Botany 10: 325, 1869. [MB#199893]

≡*Cystidiodontia laminifera* (Berk. & M.A. Curtis) Hjortstam, Mycotaxon 39: 416, 1990. [MB#127328]

=*Hydnum artocreas* Berk. & M.A. Curtis ex Cooke, Grevillea 20 (93): 1, 1891. [MB#242456]

≡*Cystostereum artocreas* (Berk. & M.A. Curtis ex Cooke) Hallenb. & Ryvarden, Mycotaxon 2 (1): 135, 1975. [MB#312562]

≡*Cystidiodontia artocreas* (Berk. & M.A. Curtis ex Cooke) Hjortstam, Mycotaxon 17: 571, 1983. [MB#108828]

Description and illustration—Hjortstam and Ryvarden [29], as *Cystidiodontia atrocreas*.

Specimens examined—Brazil, Paraná, Fazenda São Pedro, General Carneivo, on the dead trunk of a dicotyledonous tree, 10 April 1989, A.A.R. de Meijer 1383 (CFMR). Costa Rica, San José Province, Cinco Esquinas, elev. 4700 feet, on hardwood, 24 June 1963, J.L. Lowe 12856 (CFMR). Jamaica, Surrey, Saint Andrew Parish, Cinchona Botanical Gardens, along the trail to Morce’s Gap, on hardwood, 12 June 1999, K.K. Nakasone, FP-150344 (CFMR). U.S. Virgin Island, St John, Virgin Islands National Park, Bordeaux Mountain Trail, on dead hardwood branch, 9 December 1997, K.K. Nakasone, FP-150137 (CFMR).

Habitat and distribution—on dead trunks and branches of angiosperms, reported from the Caribbean Islands and Central and South America.

Notes—Hjortstam [30] created *Cystidiodontia* for *Hydnum artocreas*, but later, he determined it was a synonym of *Hydnum laminiferum* [31]. Thereafter, *Cystidiodontia laminifera* was adopted and reported from South and Central America [31], Taiwan [28], and mainland China [32]. A sample from Nigeria (TF10, ITS and LSU GenBank accession number MH625705) formed a lineage with one from Costa Rica (KHL13057, ITS and LSU GenBank accession number EU118622), thus extending the distribution of *C. laminiferus* into Africa. The reports of *Crustomyces laminiferus* from Asia [28,32] need to be confirmed, for they may be *C. isabellinus* (see discussion above).

***Crustomyces tephroleucus*** (J. Song, Y.C. Dai & B.K. Cui) Yue Li, Nakasone & S.H. He, **combinatio nova**

MycoBank: MB844201

≡*Rigidotubus tephroleucus* J. Song, Y.C. Dai & B.K. Cui, Phytotaxa 333(2): 263, 2018. [MB823703]

Specimens examined—China, Hainan Province, Baoting County, Qixianling Forest Park, on angiosperm stump, 20 November 2015, Cui 13717 (BJFC 028583, holotype); Qiongzhong County, Limushan Forest Park, on a fallen angiosperm trunk, 18 November 2015, Cui 13653 (BJFC 028519, paratype).

Habitat and distribution—on dead trunks and stumps of woody angiosperms from subtropical China.

Notes—Song et al. [5] erected the monotypic genus *Rigidotubus* based on molecular evidence and the poroid hymenophore configuration of *R. tephroleucus*. Figure 1 and Figure 2 show that *Crustomyces* (*Rigidotubus*) *tephroleucus* nested within the *Crustomyces* clade and is sister to the *C. isabellinus* and *C. laminiferus* lineage. Despite its unusual poroid hymenophore configuration, *Rigidotubus* is placed in synonymy under *Crustomyces* based on phylogenetic evidence (Figure 1 and Figure 2).

***Cystostereum crassisporum*** Yue Li, Nakasone & S.H. He, **sp. nov.** Figure 5 and Figure 6

MycoBank: MB844196

Type—China, Guangxi Autonomous Region, Huanjiang County, Mulun Nature Reserve, on dead Ficus tree, 10 July 2017, He 4740 (BJFC 024259, holotype, CFMR, isotype).

Etymology—refers to the thick-walled basidiospores.

Fruiting body—Basidiomes annual, resupinate, widely effused, closely adnate, first as small patches then confluent, up to 7 cm long, 3 cm wide, up to 350 µm thick in section, coriaceous to suberose. Hymenophore smooth to irregularly tuberculate, pale yellow (4A3) to greyish orange (5B3), without cracks; margin adnate, abrupt, distinct or rapidly thinning out, paler or concolorous with hymenophore surface. Context cream.

Microscopic structures—Hyphal system dimitic; generative hyphae bearing clamp connections and microbinding hyphae in a substrate, scarce to locally abundant. Subiculum not observed. Subhymenium composing most of context, thickening, up to 235 µm thick, a densely agglutinated tissue of hyphae and cystidia with two distinct layers—next to substrate with collapsed gloeocystidia often containing dark yellow, resinous-like material and an upper layer with elongate gloeocystidia enclosed by a thin, agglutinated matrix of hyphae, often stratose; hyphae thin-walled, colorless, smooth, interwoven, moderately septate, 1.5–2.5 µm in diam; gloeocystidia embedded, vesicular to subfusiform, sometimes with a distinct, short stalk, usually with a constriction in the upper part, empty or filled with dark yellow, solid, resinous-like material, slightly thick-walled, colorless, smooth, 17–53 × 6–19 µm. Basidia clavate, sometimes with slight constriction, thin-walled, colorless, smooth, with four sterigmata and a basal clamp connection, 14–24 × 4–6 µm. Basidiospores ellipsoid to broadly ellipsoid, with a distinct apiculus, thick-walled, colorless, smooth, usually with oil-drops, often 2–4 in groups, IKI–, CB+, (3.7–) 4–4.8 (–5.1) × (2.8–) 3–3.5 (–4) µm, L = 4.2 µm, W = 3.2 µm, Q = 1.3 (n = 60/2).

Additional specimens examined—China, Hainan Province, Baoting County, Qixianling Forest Park, on angiosperm stump, 11 June 2016, He 3995 (BJFC 022497, CFMR).

Habitat and distribution—on *Ficus* or other angiosperm trees from subtropical China.

Notes—*Cystostereum crassisporum* is characterized by thin basidiomes with a smooth to slightly tuberculate hymenophore, a dimitic hyphal system, abundant gloeocystidia, and thick-walled, cyanophilous basidiospores. The species is easily mistaken to be monomitic since microbinding hyphae are difficult to find. *Cystostereum submurrayi*, described below, is also from China but has a distinctly tuberculate hymenophore and thin-walled basidiospores. *Cystostereum kenyense* Hjortstam has thick-walled basidiospores as in *C. crassisporum* but is characterized by a tuberculate to odontoid hymenophore, a monomitic hyphal system, and produces two kinds of cystidia [33]. *Cystostereum sirmaurense* R. Kaur, Avn. P. Singh & Dhingra from India is distinguished from *C. crassisporum* by its smooth hymenophore with minute, scattered tubercules, a dimitic hyphal system with skeletal hyphae, and smaller, thin-walled basidiospores [34]. In the phylogenetic trees, *C. crassisporum* formed a distinct lineage sister to the *C. murrayi* (Berk. & M.A. Curtis) Pouzar and *C. submurrayi* clade.

***Cystostereum submurrayi*** Yue Li, Nakasone, C.C. Chen & S.H. He, **sp. nov.** Figure 7 and Figure 8

MycoBank: MB844197

Type—China, Hunan Province, Zhangjiajie, Zhangjiajie Forest Park, on a fallen angiosperm trunk, 7 July 2015, He 2301 (BJFC 020755, holotype, CFMR, isotype).

Etymology—refers to the morphological similarity and close phylogenetic relationship to *C. murrayi*.

Fruiting body—Basidiomes perennial, resupinate, widely effused, closely adnate, first as small patches, later confluent up to 12 cm long, 7 cm wide, up to 400 µm thick in section, woody hard to corneus. Hymenophore tuberculate, greyish yellow [4B (3–4)], with scattered cracks in mature areas; margin adnate, distinct, rapidly thinning out, white at first, then concolorous with hymenophore. Context greyish yellow [4B (3–4)].

Microscopic structures—Hyphal system dimitic; generative hyphae bearing clamp connections and microbinding hyphae observed in substrate. Subiculum not observed. Subhymenium composing most of context, thickening, up to 380 μm thick, structure as described for *C. crassisporum*; gloeocystidia abundant, embedded, ovoid, vesicular or subclavate, usually with long stalk when originating in the subiculum, typically devoid of contents, rarely with phloxine-stained contents or dark yellow, solid, resinous-like material, slightly thick-walled, smooth, sometimes containing oil-drops, embedded, 15–80 × 5–20 µm. Basidia clavate, thin-walled, colorless, smooth, with four sterigmata and a basal clamp connection, 12–25 × 4–5 µm. Basidiospores narrowly ellipsoid to ellipsoid, with a small, distinct apiculus, thin-walled, colorless, usually agglutinated in groups of 2–4, smooth, IKI–, CB+, 3–4.8 (–5) × 2–2.7 (–3) µm, L = 3.9 µm, W = 2.3 µm, Q = 1.6–1.8 (n = 120/4).

Additional specimens examined—China, Jiangxi Province, Lianping County, Jiulianshan Nature Reserve, on a dead angiosperm branch, 14 August 2016, He 4379 (BJFC 023820, CFMR). Taiwan, Nantou County, Jenai Township, Aowanda National Forest Recreation Area, on an angiosperm trunk, 28 April 2018, WEI 18-031 (TNM F33194) & WEI 18-033 (TNM F33196); on an angiosperm branch, 28 August 2017, WEI 17-618 (TNM F33165); on an angiosperm branch, 28 August 2017, GC 1708-337 (TNM F32300); near Bird-watching platform, on a fallen angiosperm trunk, 12 September 2018, WEI 18-405 (TNM F34221) & WEI 18-406 (TNM F34222); Hualian County, Xiulin Township, Paiyang Hiking Trail, on angiosperm wood, 10 September 2001, Wu 0109-32 (TNM F13777); Pingtung County, Manzhou Township, Chufengshan, on a fallen angiosperm trunk, 29 November 2018, WEI 18-603 (TNM F34697), WEI 18-604 (TNM F34698), WEI 18-605 (TNM F34699), WEI 18-633 (TNM F34720), WEI 18-644 (TNM F34725), WEI 18-670 (TNM F34738).

Habitat and distribution—on dead logs and branches of woody angiosperms in southeastern China and Taiwan.

Notes—*Cystostereum submurrayi* is characterized by a tuberculate hymenophore, a dimitic hyphal system, embedded gloeocystidia, and thin-walled basidiospores. Its structure is like that in *C. crassisporum*, also from China, that differs by developing slightly broader, cyanophilous, thick-walled basidiospores and a smoother hymenophore. *Cystostereum murrayi* has significantly thicker basidiomes (up to 1 mm thick) and is reported from North America and Europe. The report of *C. murrayi* in China by Dai [32] needs to be confirmed. In our phylogenetic analyses, *C. submurrayi* formed a distinct lineage sister to *C. murrayi*, which appears to have two distinct lineages in Europe (Figure 1 and Figure 2). *Cystostereum sirmaurense* also has thin-walled basidiospores as in *C. submurrayi* but develops skeletal hyphae, smaller gloeocystidia (26–44 × 7–8.5 µm), and a mostly smooth hymenophore [34]).

***Effusomyces*** Yue Li, Nakasone & S.H. He, **gen. nov.**

MycoBank: MB844194

Type species—*Effusomyces thailandicus* Yue Li, Nakasone & S.H. He

Etymology—refers to the broadly effused fruitbody.

Basidiomes annual, resupinate, effused, adnate, membranaceous. Hymenial surface smooth, pale yellow to greyish yellow, unchanged in KOH. Hyphal system monomitic; generative hyphae clamped. Subiculum thin; hyphae slightly thick-walled, colorless, smooth. Subhymenium indistinct. Hyphidia rare. Cystidia or gloeocystidia absent. Basidia clavate to subcylindrical, thin-walled, with four sterigmata and a basal clamp connection. Basidiospores ellipsoid, thin-walled, colorless, smooth, IKI–, CB–. Associated with a white rot of dead hardwood branches and bamboo.

Notes—*Effusomyces* are characterized by thin, membranous basidiomes, a smooth hymenophore, a monomitic hyphal system with clamped generative hyphae, and a lack of cystidia or gloeocystidia. Although lacking distinctive morphological features, *Effusomyces* are nested within the Cystostereaceae and are sister to *Crustomyces* (Figure 1 and Figure 2). *Crustomyces* (*Rigidotubus*) *tephroleucus* also is monomitic and lacks cystidia and gloeocystidia but has a tough, thick (up to 0.5 mm) basidiome and poroid hymenophore. Within the family, monomitic hyphal systems with clamped generative hyphae are also reported in *Parvobasidium*, *Parvodontia*, and some species of *Crustomyes* and *Cystostereum*. Still, these taxa also have cystidia, gloeocystidia, or thick-walled basidiospores.

***Effusomyces thailandicus*** Yue Li, Nakasone & S.H. He, **sp. nov.** Figure 9

MycoBank: MBMB844195

Type—Thailand, Chiang Rai, Campus of Mae Fah Luang University, on dead bamboo, 21 July 2016, He 4055c (BJFC 023496, holotype, CFMR, isotype).

Etymology—refers to Thailand, the type locality.

Fruiting body—Basidiomes annual, resupinate, widely effused, adnate, inseparable from substrate, membranaceous, first as small patches, later confluent up to 12 cm long, 3 cm wide, up to 20 µm thick in section. Hymenophore smooth, pale yellow (4A3) or greyish yellow (4B3), unchanged in KOH, without cracks; margin thinning out, adnate, indistinct, concolorous with hymenophore surface. Context cream-colored.

Microscopic structures—Hyphal system monomitic; generative hyphae bearing clamp connections. Subiculum thin, sparse, dense tissue with hyphae arranged more or less parallel to substrate; hyphae slightly thick-walled, colorless, smooth, infrequently branched, moderately septate, 2.5–3 µm in diameter Subhymenium indistinct, a sparsely interwoven tissue; hyphae thin-walled, colorless, smooth, infrequently branched, moderately septate, 1–2 µm in diameter Hyphidia rare, simple, filamentous or with lateral or apical bumps or knobs, thin-walled, colorless, smooth, 17–31 × 1.5–2 µm. Cystidia or gloeocystidia absent. Basidia clavate to subcylindrical, usually with a slight medial constriction, thin-walled, colorless, smooth, with four sterigmata and a basal clamp connection, 10–26 × 3–6 µm. Basidiospores ellipsoid, with tiny apiculus, thin-walled, colorless, smooth, IKI–, CB–, 5–6.5 × 3–4.2 (–4.9) µm, L = 5.9 µm, W = 3.8 µm, Q = 1.5–1.6 (n = 90/3).

Additional specimens examined—Thailand, Chiang Rai, Campus of Mae Fah Luang University, on a dead angiosperm branch, 21 July 2016, He 4041 (BJFC 023480, CFMR); Mork Fae Waterfall, on dead bamboo, 25 July 2016, He 4126 (BJFC 023568, CFMR).

Habitat and distribution—on culm of dead bamboo or angiosperm branch from Thailand.

Notes—Except for the small basidia producing relatively large basidiospores and very thin basidiomes, there are no outstanding morphological features of *E. thailandicus*. The hyphidia are easily overlooked, for they are inconspicuous and scarce.

***Parvodontia austrosinensis*** Yue Li, Nakasone & S.H. He, **sp. nov.** Figure 10

MycoBank: MB846254

Diagnosis—Similar to *P. luteocystidia* and *P. albocrustacea* except with thinner basidiomes, up to 70 µm thick, smooth hymenophore with scattered, small, rounded tubercules and microbinding hyphae.

Type—China, Guangxi Autonomous region, Huanjiang County, Mulun Nature Reserve, on a dead angiosperm trunk, 10 July 2017, He 4730 (BJFC 024249, holotype; CFMR, isotype).

Etymology—refers to the type locality in southern China.

Fruiting body—Basidiomes annual, resupinate, widely effused, first as small patches, later confluent up to 25 cm long, 2.5 cm wide, adnate, thin, up to 70 µm thick in section, pellicular. Hymenophore smooth with scattered, inconspicuous, rounded tubercules, greyish yellow (4B4) or greyish orange (6B4), darkening in KOH, with scattered cracks; margin thinning out, adnate, indistinct, concolorous with hymenophore surface. Context greyish yellow (4B4).

Microscopic structures—Hyphal system dimitic; generative hyphae bearing clamp connections; microbinding hyphae abundant in context. Subiculum is an open, loosely interwoven tissue of subicular hyphae, microbinding hyphae, and embedded vesicles; subicular hyphae are thin to slightly thick-walled, colorless, smooth, (1.5–) 2.2–5 µm in diam; microbinding hyphae abundant, thin-walled, colorless, smooth, frequently branched, often at right angles, aseptate, 0.7–2 µm in diam, lacking a lumen; vesicles numerous, spherical sometimes with digitiform appendages, often with short stalk or subfusiform, terminal or intercalary, usually empty, rarely with honey-yellow contents, walls slightly thick, colorless, smooth, often weakly cyanophilous, 10–18 µm in diameter Subhymenium is a somewhat densely interwoven tissue composed of subhymenial and microbinding hyphae and embedded gloeocystidia; subhymenial hyphae are thin- to slightly thick-walled, colorless, smooth, infrequently branched, frequently septate with clamps, 2–4 µm in diam; microbinding hyphae as described above; gloeocystidia abundant, vesicular as above or irregular in shape from fusiform to clavate, usually with multiple short appendages or lobes, containing resinous-like, dark honey yellow material that is darker yellow in Melzer’s reagent, walls thin-walled, colorless, smooth, 12–25 × (4–)7–10 µm. Hymenium of cystidia, gloeocystidia, and basidia. Cystidia rare, inconspicuous, subfusiform, with a basal clamp connection, contents staining evenly in phloxine, walls thin, colorless, smooth, 6–9 × 3 µm. Hymenial gloeocystidia spherical to ovate with short stalk, typically with one or more digitiform appendages, containing resinous-like, dark honey yellow material, walls slightly thick-walled, colorless, smooth, 11–27 × 10–25 µm. Basidia clavate with a slightly median constriction, thin-walled, colorless, smooth, with four slender sterigmata and a basal clamp connection, 6–13 × 3–5 µm. Basidiospores ellipsoid to subcylindrical, thin-walled, colorless, smooth, IKI–, CB–, (3.7–) 4–5.3 (–5.5) × 2.3–3 (–3.2) µm, L = 4.8 µm, W = 2.7 µm, Q = 1.7–1.8 (n = 60/2).

Additional specimens examined—China, Jiangxi Province, Lianping County, Jiulianshan Nature Reserve, on a dead angiosperm branch, 13 August 2016, He 4339 (BJFC 023781, CFMR).

Habitat and distribution—on bark and wood of dead angiosperm trunks and branches in southeastern China.

Notes—*Parvodontia austrosinensis* is characterized by thin, grayish yellow to grayish orange, smooth basidiomes with scattered, small tubercules, microbinding hyphae, and irregularly-shaped or vesicular gloeocystidia with one or more lobes or appendages. It is the first report of this genus in Asia, and the pellicular nature of basidiome makes it difficult to study its structure. *Parvodontia luteocystidia* Hjortstam & Ryvarden and *P. albocrustacea* (Rick) Baltazar & Rajchenb. are much thicker with abundant, small aculei, larger basidia (10–17 µm long) and lack microbinding hyphae. They occur on bamboo and hardwood, respectively, in South America [35,36].


**A key to genera and species in Cystostereaceae sensu lato**
1. Gloeocystidia absent………………………………………………………………………………21. Gloeocystidia present..……………………………………………………………………………32. Hymenophore poroid....………………………………………………*Crustomyces tephroleucus*2. Hymenophore smooth..………………………………………………………………*Effusomyces*3. Microbinding hyphae present...…………………………………………………………………43. Microbinding hyphae absent, skeletal hyphae present or not........…………………………64. Gloeocystidia tubular, subulate or subfusiform, encrusted...………………………*Cericium*4. Gloeocystidia varied shaped, smooth.....……………………………………………....………55. Gloeocystidia submoniliform...……………………………………………*Crustomyces albidus*5. Gloeocystidia spherical to ovate with short stalk..………………*Parvodontia austrosinensis*6. Hymenophore smooth.…..………………………………………………………………………76. Hymenophore grandinioid, odontoid or hydnoid...…………………………………………87. Basidiospores thick-walled....………………………………………*Cystostereum crassisporum*7. Basidiospores thin-walled..………………………………………………………*Parvobasidium*8. Hyphal system monomitic.….………………………………………………………*Parvodontia*8. Hyphal system dimitic..…………………………………………………………………………99. Dendrohyphidia present.…..……....………………………………………………*Crustomyces*9. Dendrohyphidia absent..…..………………………………………………………*Cystostereum*

## 4. Discussion

Cystostereaceae is a small, monophyletic family in the Agaricales, with affinities to *Lyophyllum* and allied taxa [2,3]. Phylogenetic analyses of LSU and ITS sequences confirm that *Cystostereum*, *Crustomyces*, *Cystidiodontia*, *Parvodontia*, and *Rigidotubus* belong in the Cystostereaceae. In contrast, the placement of *Cericium* and *Parvobasidium* needs to be confirmed by molecular evidence. Our phylogenetic analyses support the monophyly of the family and clarify generic and species relationships within the family (Figure 1 and Figure 2). We demonstrate that *Crustomyces* has a wide range of hymenophore configurations from smooth, odontoid, and spinose to poroid for the types species of *Crustomyces*, *Cystidiodontia*, and *Rigidotubus* all clustered together in a monophyletic clade with high support values. The newly emended *Crustomyces* includes seven species—*C. subabruptus*, the type species, *C. indecorus* Hjortstam, *C. pini-canadensis*, the new taxon *C. albidus* described herein, and three new combinations proposed above, *C. isabellinus*, *C. laminiferum*, and *C. tephroleucus*. Excluded from the genus is *Crustomyces expallens* (Bres.) Hjortstam because its large, narrowly clavate basidia are unlike the typical, shorter clavate basidia of other species in the genus. *Crustomyces heteromorphum* (Hallenb.) Hjortstam from eastern Europe is a synonym of *C. subabruptus*, as posited by Larsson and Ryvarden [24]. There are additional species of *Crustomyces* from China and New Zealand that appear as distinct lineages represented by Cui 12283 and HHB 19125, respectively, in Figure 2.

In addition to *Crustomyces* and *Cystostereum*, two lineages, *Parvodontia* and *Effusomyces*, were revealed in the phylogenetic analyses (Figure 1 and Figure 2). *Parvodontia* is similar to other species in the Cystostereaceae that it produces vesicles and gloeocystidia. Effusomyces is an unlikely member because it has a very simple basidiome structure and lacks vesicles and gloeocystidia.

Once again, molecular phylogenetic analyses show that hymenophore morphology can be plastic and have varied generic features in the crust fungi. *Xylodon* (Pers.) Gray, as in *Crustomyces*, includes species with smooth, odontoid, hydnoid, and poroid hymenophores and varied microscopic features [37]. Likewise, Chen et al. [38] included taxa in *Irpex* with effused, effused-reflexed, and stipitate-pileate basidiomes with smooth, poroid, labyrinthine, irpicoid, and hydnoid hymenophores. Another example is *Dentocorticium* (Parmasto) M.J. Larsen & Gilb, which includes taxa with varied basidiome habits as well as hymenophore configuration [39]. *Lopharia mirabilis* (Berk. & Broome) Pat. is an example of a species with an extremely varied hymenophore ranging from tuberculate, odontoid, irpicoid to subporoid [39,40,41].

Recently, many new macrofungal taxa have been described from Southern China [42,43,44,45,46,47,48], and the area is very rich in wood-inhabiting fungi [49,50,51]. In the present paper, five new species belonging to Cystostereaceae are described, including four species from Southern China. It seems more unknown macrofungal species exist in the area, and further investigations are needed to demonstrate the fungal diversity in Southern China.

## Figures and Tables

**Figure 1 jof-08-01229-f001:**
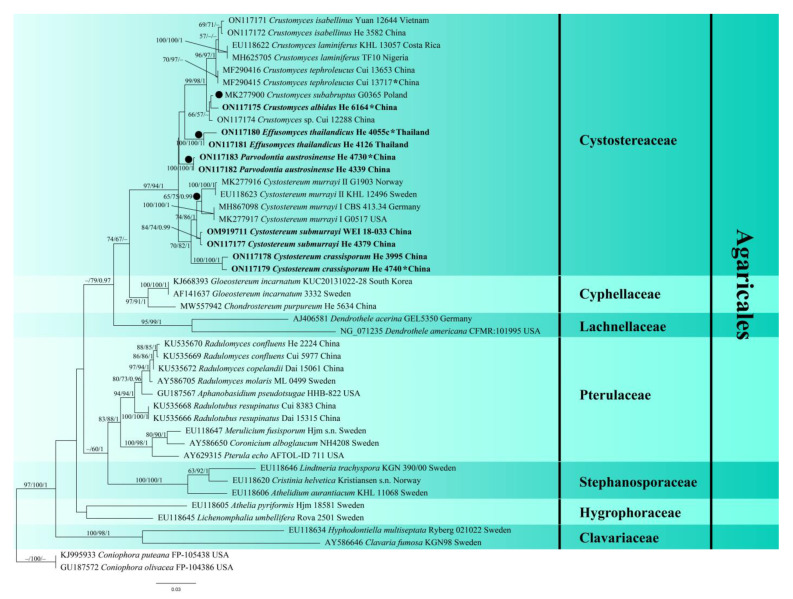
Phylogenetic tree of ML analysis from the LSU sequences of Agaricales taxa. Branches are labeled with parsimony bootstrap values (≥50%, first), likelihood bootstrap values (≥50%, second), and Bayesian posterior probabilities (≥0.95, third). New species are set in bold, type species of genera in Cystostereaceae are marked with a solid circle (●) in front of the branch, and holotypes of new taxa are marked with an asterisk (*).

**Figure 2 jof-08-01229-f002:**
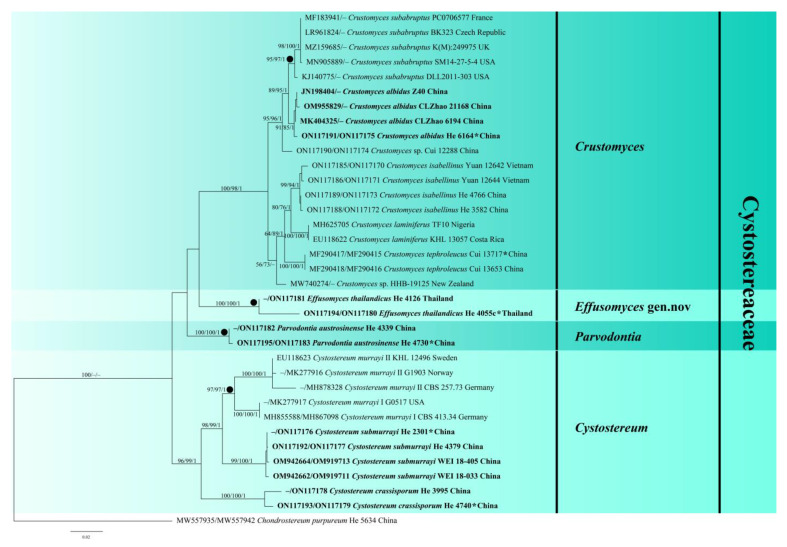
Phylogenetic tree of ML analysis from the concatenated ITS-LSU sequences of Cystostereaceae taxa. Branches are labeled with parsimony bootstrap values (≥50%, first), likelihood bootstrap values (≥50%, second), and Bayesian posterior probabilities (≥0.95, third). New species are set in bold; type species are marked with a solid circle (●) in front of the branch, and holotypes of new taxa are marked with an asterisk (*).

**Figure 3 jof-08-01229-f003:**
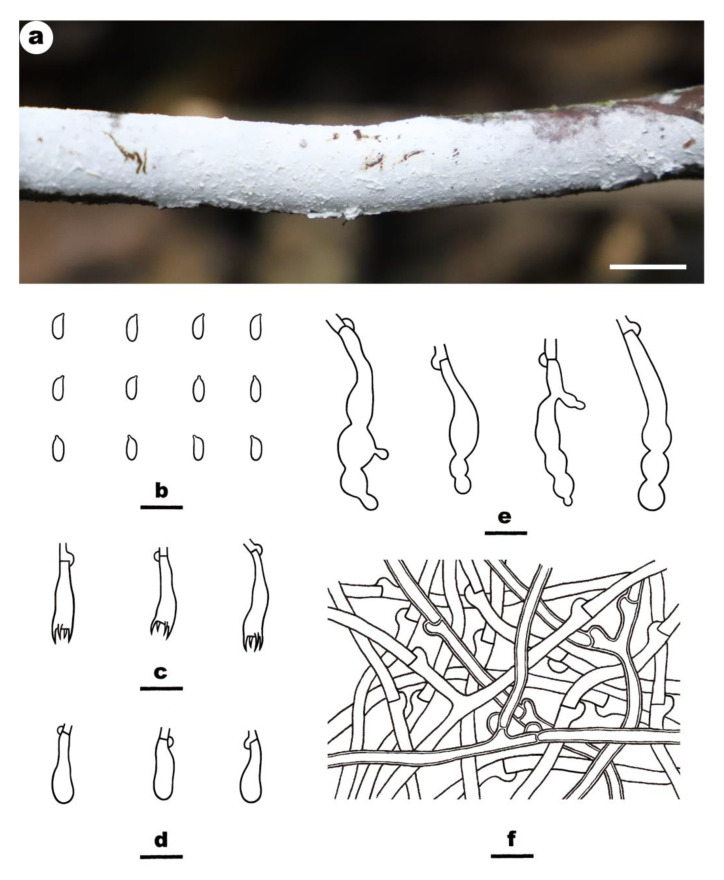
*Crustomyces albidus* (from the holotype BJFC 033109). Scale bars: (**a**) = 1 cm; (**b**–**f**) = 10 µm. (**a**). Basidiomes; (**b**). Basidiospores; (**c**). Basidia; (**d**). Basidioles; (**e**). Gloeocystidia; (**f**). Generative hyphae from subiculum.

**Figure 4 jof-08-01229-f004:**
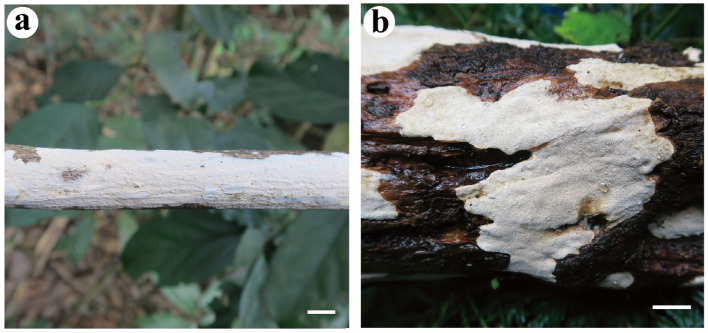
Basidiomes of *Crustomyces isabellinus*. Scale bars: (**a**–**f**) = 1 cm. (**a**). He 3582 (BJFC 022082); (**b**). He 4755 (BJFC 024274); (**c**). He 4852 (BJFC 024371); (**d**). He 4872 (BJFC 024391); (**e**). He 4884 (BJFC 024403); (**f**). He 5995 (BJFC 030831).

**Figure 5 jof-08-01229-f005:**
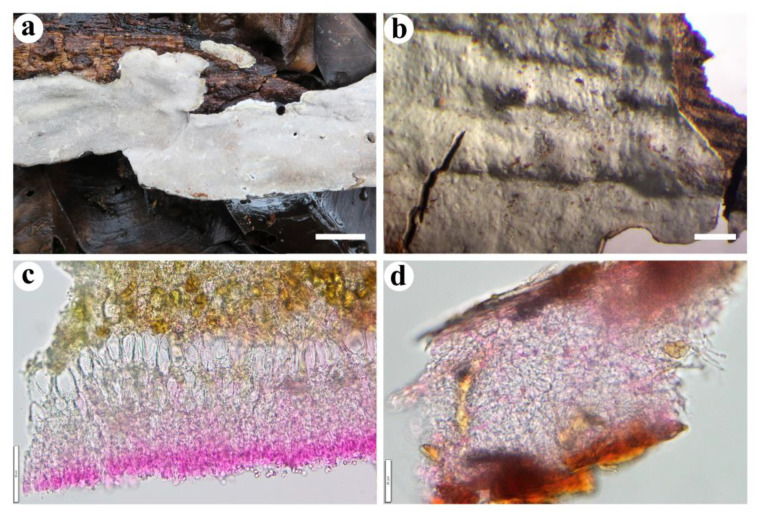
*Cystostereum crassisporum* (from the holotype BJFC 024259). Scale bars: (**a**) = 1 cm; (**b**) = 1 mm; (**c**) = 50 µm; (**d**) = 20 µm. (**a**,**b**). Basidiomes; (**c**). Part of a vertical section through basidiome; (**d**). Part of a vertical section from the substrate showing the microbinding hyphae.

**Figure 6 jof-08-01229-f006:**
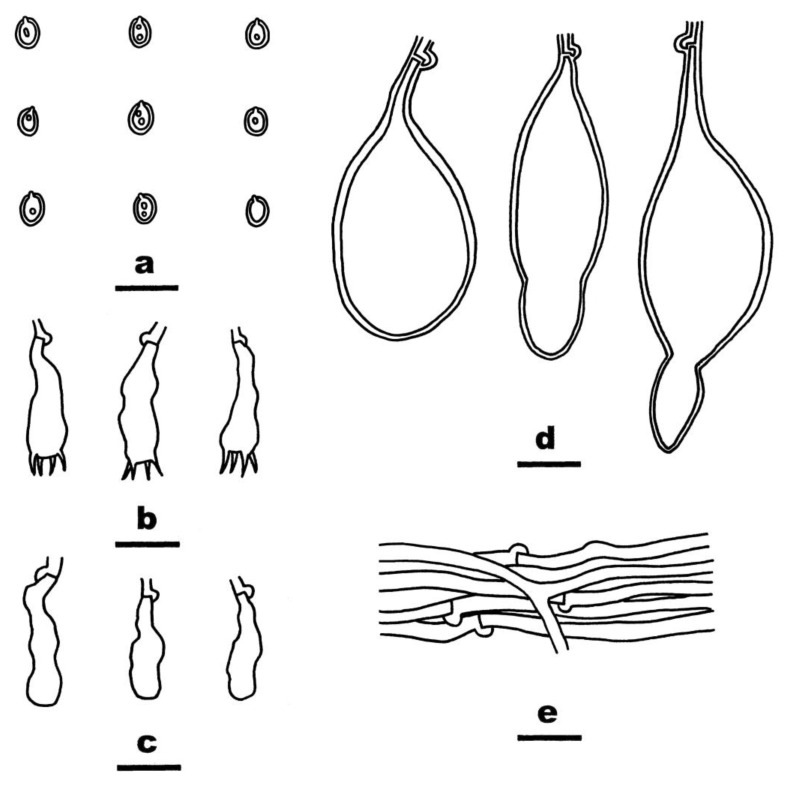
*Cystostereum crassisporum* (from the holotype BJFC 024259). Scale bars: (**a**–**e**) = 10 µm. (**a**). Basidiospores; (**b**). Basidia; (**c**). Basidioles; (**d**). Gloeocystidia; (**e**). Generative hyphae from subiculum.

**Figure 7 jof-08-01229-f007:**
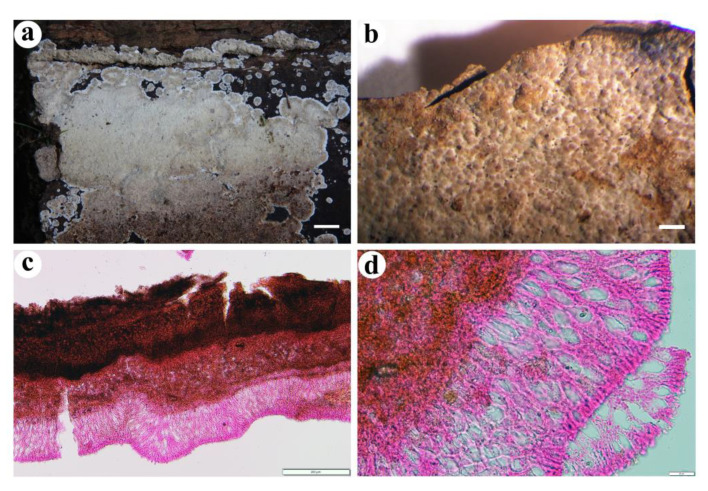
*Cystostereum submurrayi* (from the holotype BJFC 020755). Scale bars: (**a**) = 1 cm; (**b**) = 1 mm; (**c**) = 200 µm; (**d**) = 20 µm. (**a**,**b**). Basidiomes; (**c**). Part of a vertical section through basidiome; (**d**). Close-up of a vertical section through basidiome.

**Figure 8 jof-08-01229-f008:**
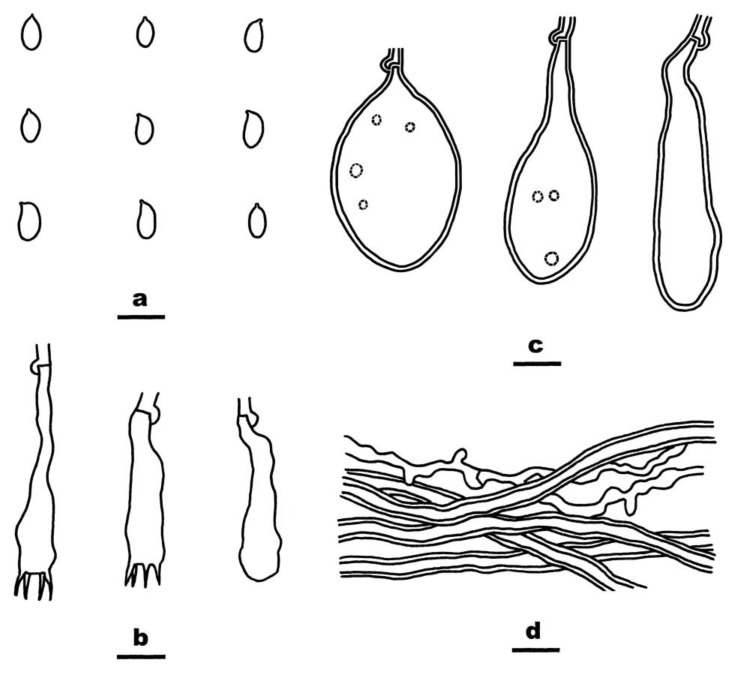
*Cystostereum submurrayi* (from the holotype BJFC 020755). Scale bars: (**a**–**d**) = 10 µm. (**a**). Basidiospores; (**b**). Basidia and a Basidiole; (**c**). Gloeocystidia; (**d**). Hyphae from subiculum.

**Figure 9 jof-08-01229-f009:**
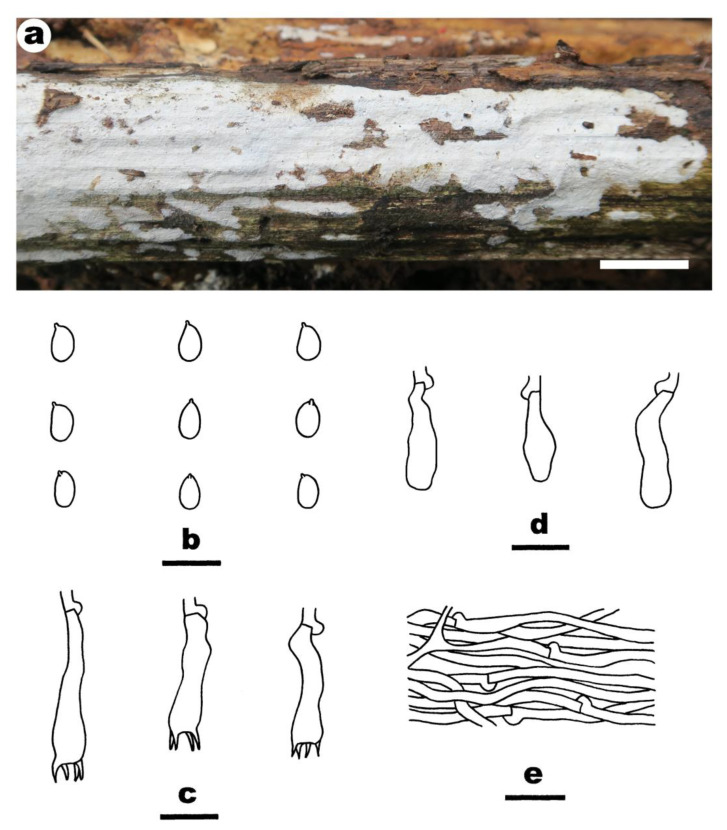
*Effusomyces thailandicus* ((**a**) from BJFC 023568; (**b**–**e**) from the holotype BJFC 023496). Scale bars: (**a**) = 1 cm; (**b**–**e**) = 10 µm. (**a**). Basidiomes; (**b**). Basidiospores; (**c**). Basidia; (**d**). Basidioles; (**e**). Hyphae from subiculum.

**Figure 10 jof-08-01229-f010:**
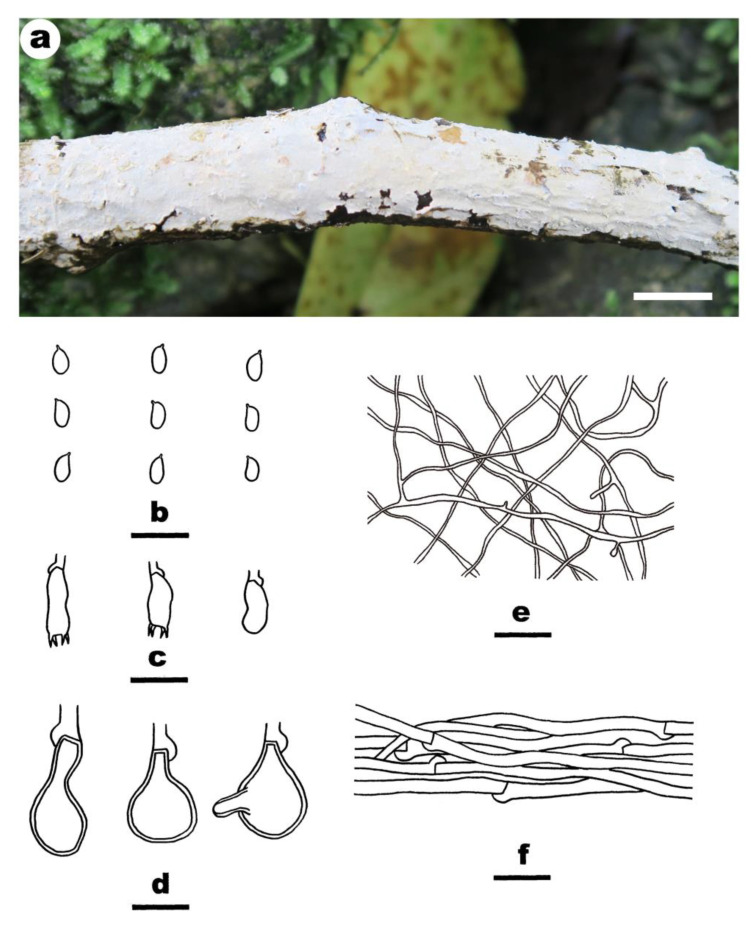
*Parvodontia austrosinensis* (from the holotype BJFC 024249). Scale bars: (**a**) = 1 cm; (**b**–**e**) = 10 µm. (**a**). Basidiomes; (**b**). Basidiospores; (**c**). Basidia and a basidiole; (**d**). Gloeocystidia; (**e**). Microbinding hyphae from subhymenium; (**f**). Generative hyphae from subiculum.

**Table 1 jof-08-01229-t001:** Species and sequences used in the phylogenetic analyses. New species and combinations are set in bold with type specimens indicated with an asterisk (*).

Taxa	Voucher	Locality	ITS	nrLSU	Reference
*Aphanobasidium pseudotsugae*	HHB-822	USA	—	GU187567	[17]
*Athelia pyriformis*	Hjm 18581	Sweden	—	EU118605	[2]
*Athelidium aurantiacum*	KHL 11068	Sweden	—	EU118606	[2]
*Chondrostereum purpureum*	He 5634	China	MW557935	MW557942	Unpublished
*Clavaria fumosa*	KGN98	Sweden	—	AY586646	[18]
*Coniophora puteana*	FP-105438	USA	—	KJ995933	Unpublished
*Coniophora olivacea*	FP-104386	USA	—	GU187572	[17]
*Coronicium alboglaucum*	NH4208	Sweden	—	AY586650	[18]
*Cristinia helvetica*	Kristiansen s.n.	Norway	—	EU118620	[2]
** *Crustomyces albidus* **	**He 6164 ***	**China**	**ON117191**	**ON117175**	**present study**
** *Crustomyces albidus* **	**CLZhao 6194**	**China**	**MK404325**	**—**	**Unpublished**
** *Crustomyces albidus* **	**CLZhao 21168**	**China**	**OM955829**	**—**	**Unpublished**
** *Crustomyces albidus* **	**Z40**	**China**	**JN198404**	**—**	**[19]**
** *Crustomyces isabellinus* **	**Yuan 12642**	**Vietnam**	**ON117185**	**ON117170**	**present study**
** *Crustomyces isabellinus* **	**Yuan 12644**	**Vietnam**	**ON117186**	**ON117171**	**present study**
** *Crustomyces isabellinus* **	**He 3582**	**China**	**ON117188**	**ON117172**	**present study**
** *Crustomyces isabellinus* **	**He 4766**	**China**	**ON117189**	**ON117173**	**present study**
** *Crustomyces laminiferus* **	**KHL 13057**	**Costa Rica**	**EU118622**	**EU118622**	**[2]**
** *Crustomyces laminiferus* **	**TF10**	**Nigeria**	**MH625705**	**MH625705**	**Unpublished**
*Crustomyces* sp	Cui 12288	China	ON117190	ON117174	present study
*Crustomyces* sp	HHB-19125	New Zealand	MW740274	—	Unpublished
*Crustomyces subabruptus*	DLL2011-303	USA	KJ140775	—	Unpublished
*Crustomyces subabruptus*	BK323	Czech Republic	LR961824	—	Unpublished
*Crustomyces subabruptus*	K(M):249975	UK	MZ159685	—	Unpublished
*Crustomyces subabruptus*	PC0706577	France	MF183941	—	Unpublished
*Crustomyces subabruptus*	SM14-27-5-4	USA	MN905889	—	Unpublished
*Crustomyces subabruptus*	G0365	Poland	—	MK277900	[20]
** *Crustomyces tephroleucus* **	**Cui 13653**	**China**	**MF290418**	**MF290416**	**[5]**
** *Crustomyces tephroleucus* **	**Cui 13717 ***	**China**	**MF290417**	**MF290415**	**[5]**
*Cystostereum murrayi*Ⅰ	CBS 413.34	Germany	MH855588	MH867098	[21]
*Cystostereum murrayi*Ⅰ	G0517	USA	—	MK277917	[20]
*Cystostereum murrayi*Ⅱ	KHL 12496	Sweden	EU118623	EU118623	[2]
*Cystostereum murrayi*Ⅱ	G1903	Norway	—	MK277916	[20]
*Cystostereum murrayi*Ⅱ	CBS 257.73	Germany	—	MH878328	[21]
** *Cystostereum crassisporum* **	**He 3995**	**China**	**—**	**ON117178**	**present study**
** *Cystostereum crassisporum* **	**He 4740 ***	**China**	**ON117193**	**ON117179**	**present study**
** *Cystostereum submurrayi* **	**He 2301 ***	**China**	**—**	**ON117176**	**present study**
** *Cystostereum submurrayi* **	**He 4379**	**China**	**ON117192**	**ON117177**	**present study**
** *Cystostereum submurrayi* **	**WEI 18-033**	**China**	**OM942662**	**OM919711**	**present study**
** *Cystostereum submurrayi* **	**WEI 18-405**	**China**	**OM942664**	**OM919713**	**present study**
*Dendrothele acerina*	GEL5350	Germany	—	AJ406581	Unpublished
*Dendrothele americana*	CFMR:101995	USA	—	NG_071235	[22]
** *Effusomyces thailandicus* **	**He 4055c ***	**Thailand**	**ON117194**	**ON117180**	**present study**
** *Effusomyces thailandicus* **	**He 4126**	**Thailand**	**—**	**ON117181**	**present study**
*Gloeostereum incarnatum*	KUC20131022-28	South Korea	—	KJ668393	Unpublished
*Gloeostereum incarnatum*	3332	Sweden	—	AF141637	Unpublished
*Hyphodontiella multiseptata*	Ryberg 021022	Sweden	—	EU118634	[2]
*Lichenomphalia umbellifera*	Rova 2501	Sweden	—	EU118645	[2]
*Lindtneria trachyspora*	KGN 390/00	Sweden	—	EU118646	[2]
*Merulicium fusisporum*	Hjm s.n.	Sweden	—	EU118647	[2]
** *Parvodontia austrosinense* **	**He 4339**	**China**	**—**	**ON117182**	**present study**
** *Parvodontia austrosinense* **	**He 4730 ***	**China**	**ON117195**	**ON117183**	**present study**
*Pterula echo*	AFTOL-ID 711	USA	—	AY629315	Unpublished
*Radulomyces confluens*	Cui 5977	China	—	KU535669	[23]
*Radulomyces confluens*	He 2224	China	—	KU535670	[23]
*Radulomyces copelandii*	Dai 15061	China	—	KU535672	[23]
*Radulomyces molaris*	ML 0499	Sweden	—	AY586705	[18]
*Radulotubus resupinatus*	Cui 8383	China	—	KU535668	[23]
*Radulotubus resupinatus*	Dai 15315	China	—	KU535666	[23]

## Data Availability

Not applicable.

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
