# Peer review of "Taxonomy and Phylogeny of Cystostereaceae (Agaricales, Basidiomycota): A New Genus, Five New Species, and Three New Combinations"

_jof, 2022, doi:10.3390/jof8111229_

Round 1
Reviewer 1 Report
Review of the paper titled " Taxonomy and phylogeny of Cystostereaceae (Agaricales, Basidiomycota): a new genus, five new species and three new combinations"
The paper is original; it was well-written and well-organized. It reads well and is of importance to fungal taxonomists particularly those working in order Agaricales.
I would recommend publication in the JoF following minor revisions by the authors:
- The English language should be improved, particularly in punctuation and commas (Please see the comments on the attached file)
- The abstract should include at the beginning, the aim of the work and end with recommendations

Author Response
Point 1: The English language should be improved, particularly in punctuation and commas (Please see the comments on the attached file)
Response 1: The authors accepted all the corrections of the language annotated in the attached file.
Point 2: The abstract should include at the beginning, the aim of the work and end with recommendations
Response 2: The authors added the aim at the beginning of the abstract as follows: “The aim of this paper is to understand the species diversity, taxonomy and phylogeny of Cystostereaceae (Agaricales), which is based primarily on material from East and Southeast Asia.”, and the recommendations at the end of the abstract as follows: “Our results proved that the species diversity of wood-decaying fungi in East and Southeast Asia is rich, and suggested that more investigations and studies should be carried out in future.”.
Reviewer 2 Report
The work is a very important contribution to the phylogeny and taxonomy of the Cystostereaceae family, as well as new species described.
Although it appears to be basic taxonomic information, it is very useful even for molecular identification.
I would suggest that in the future the described species can be sequenced.
Author Response
Point 1: I would suggest that in the future the described species can be sequenced.
Response 1: Yes, we sequenced every specimen of the described species in the paper and will provide sequences for the new species we describe in the future.
Reviewer 3 Report
The present work is focused on a new genus, five new species and three new combinations of the family Cystostereaceae from East Asia and Southeast Asia based on morphological and molecular data. This manuscript is well presented and suitable to publish in the Journal of Fungi. Some corrections or queries are made in the pdf file.
1. In the abstract: the author said materials from East Asia were undertaken. However, samples from Thailand are also included and this material provides good information for this article as it represents a new genus and type species of Effusomyces. This sample does not belong to East Asia. Pls check it.
2. Author merged Cystidiodontia and Rigidotubus to Crustomyces and made new combinations. What are the synapomorphic traits among them?
3. Two collections of the newly proposed genus Effusomyces have some genetic/intra variation based on Figure 2. I think you should talk a bit about variation among the collections.
4. New genus can be placed just after the taxonomy and then other species.
5.
Therefore, this manuscript can be accepted after minor revision.

Author Response
Point 1: In the abstract: the author said materials from East Asia were undertaken. However, samples from Thailand are also included and this material provides good information for this article as it represents a new genus and type species of Effusomyces. This sample does not belong to East Asia. Pls check it.
Response 1: Yes, some specimens were from Vietnam and Thailand, which belongs to Southeast Asia. The authors changed “East Asia” to “East and Southeast Asia”.
Point 2: Author merged Cystidiodontia and Rigidotubus to Crustomyces and made new combinations. What are the synapomorphic traits among them?
Response 2: The authors made the new combinations based mainly on the phylogenetic analyses, but there are important synapomorphic traits among species of the three genera: a dimitic hyphal system, the presence of gloeocystidia and thin-walled, inamyloid small basidiospores.
Point 3: Two collections of the newly proposed genus Effusomyces have some genetic/intra variation based on Figure 2. I think you should talk a bit about variation among the collections.
Response 3: The authors believe that the variation of Effusomyces thailandicus on Figure 2 is acceptable. It is because that the ITS sequence of specimen He4126 is unavailable. The authors confirmed that there are no distinct morphological differences between the two specimens.
Point 4: New genus can be placed just after the taxonomy and then other species.
Response 4: Yes, we tried to arrange the “taxonomy” part as: new genus, new species and then new combinations. However, for Crustomyces, there are new species and new combinations. Thus, we arranged in alphabetical order of the genera. The authors retain the order in avoid of lots of changes of the “taxonomy” and “references” parts.
Point 5: Therefore, this manuscript can be accepted after minor revision.
Response 5: The authors revised the manuscript according to all the comments and suggestions.
Point 6: In the figure legends part: keep the herbarium number rather than collector number
Response 6: The authors replaced all the collector numbers with herbarium numbers in all the figure legends.
Point 7: In the notes part of Crustomyces laminiferus: write the sequence name.
Response 7: The authors added “ITS and LSU” for the name, because the sequences contains both ITS and LSU regions.
Point 8: In the discussion part: one new or two new lineages here? Parvodontia is an existing genus.
Response 8: Parvodontia is an existing genus but had not been sequenced and subjected to phylogenetic analyses before the present study. The authors changed “new lineages” to “lineages” in avoid of misunderstanding of readers.